# Meteorological and Nutrient Conditions Influence Microcystin Congeners in Freshwaters

**DOI:** 10.3390/toxins11110620

**Published:** 2019-10-26

**Authors:** Zofia E. Taranu, Frances R. Pick, Irena F. Creed, Arthur Zastepa, Sue B. Watson

**Affiliations:** 1Department of Biology, University of Ottawa, Ottawa K1N 6N5, ON, Canada; Frances.Pick@uottawa.ca; 2Aquatic Contaminants Research Division, Environment and Climate Change Canada, Montreal H2Y 2E7, QC, Canada; 3School of Environment and Sustainability, University of Saskatchewan, Saskatoon S7N 5C8, SK, Canada; icreed@uwo.ca; 4Canada Centre for Inland Waters, Environment and Climate Change Canada, Burlington, ON L7S 1A1, Canada; arthur.zastepa@canada.ca; 5Department of Biology, University of Waterloo, Waterloo, ON N2L 3G1, Canada; jkswatson@shaw.ca

**Keywords:** Cyanotoxins, microcystin congeners, MC-LA, nutrients, climate, Great Lakes, raw water intake, multivariate statistics, long-term monitoring

## Abstract

Cyanobacterial blooms increasingly impair inland waters, with the potential for a concurrent increase in cyanotoxins that have been linked to animal and human mortalities. Microcystins (MCs) are among the most commonly detected cyanotoxins, but little is known about the distribution of different MC congeners despite large differences in their biomagnification, persistence, and toxicity. Using raw-water intake data from sites around the Great Lakes basin, we applied multivariate canonical analyses and regression tree analyses to identify how different congeners (MC-LA, -LR, -RR, and -YR) varied with changes in meteorological and nutrient conditions over time (10 years) and space (longitude range: 77°2′60 to 94°29′23 W). We found that MC-LR was associated with strong winds, warm temperatures, and nutrient-rich conditions, whereas the equally toxic yet less commonly studied MC-LA tended to dominate under intermediate winds, wetter, and nutrient-poor conditions. A global synthesis of lake data in the peer-reviewed literature showed that the composition of MC congeners differs among regions, with MC-LA more commonly reported in North America than Europe. Global patterns of MC congeners tended to vary with lake nutrient conditions and lake morphometry. Ultimately, knowledge of the environmental factors leading to the formation of different MC congeners in freshwaters is necessary to assess the duration and degree of toxin exposure under future global change.

## 1. Introduction

The incidence of severe cyanobacterial blooms is increasing worldwide [1,2,3,4,5], along with the risk of exposure to cyanobacterial toxins [6,7,8]. Cyanotoxins are found on all continents [9], though among the suite of cyanotoxins that occur in freshwater ecosystems, microcystins (MCs) are the most commonly reported [9] and the most diverse, with over 240 different structural variants (congeners) identified to date [10]. Despite considerable research on MCs, little is known about the global or multi-region occurrence of the different congeners, and how their distribution relates to environmental conditions (though see key work by [11,12]). This inability to predict the spatial (and temporal) variability in MC congeners is a major concern given that numerous wildlife and livestock fatalities have been linked to exposure to these cyanotoxins [13,14]. Humans are also at risk. People living in close proximity to lakes with frequent MC-producing cyanobacterial blooms or exposed to contaminated drinking or recreational water have been reported to experience various health problems such as muscle pain and gastrointestinal, skin and ear irritations [15]. There is also mounting evidence of chronic health problems associated with MC exposure, including a higher incidence of non-alcoholic liver cancer [16,17,18,19]. More recently, MCs have been shown to transfer up the pelagic food chain to fish [20], with additional long-term implications for human health [21,22]. 

Although new MC congeners are continually being identified [10], only a small number are monitored on a regular basis, largely due to the lack of standards and the need to use advanced LC/MS/MS techniques to identify multiple congeners. More routinely, total MC concentrations are reported as MC-LR equivalents based on enzyme-linked immunosorbent assays (ELISA) or protein phosphatase inhibition bioassays (PPI). Among MC congeners, MC-LR is considered the most common and widely distributed [23,24], however, the use of ELISA kits by many agencies may create a bias as the method has a lower reactivity to MC congeners other than MC-LR. This bias would be most extreme in regions where MC-LR is not the most common congener or not among the MC congeners detected (e.g., [25,26]), leading to an underestimation of total MC concentrations. For instance, MC-RR and -YR have been increasingly reported globally (e.g., [27]), and there are reports of MC-LA dominance in Canada (e.g., in Canada [28,29]) and in the US (e.g., Midwest [24]), suggesting that these variants are more common than previously thought [30]. From a toxicity point of view, the occurrence of MC-LA dominating blooms in some North American lakes can have important consequences on animal health. MC-LA has been shown to be as toxic as MC-LR [31,32] and as persistent if not more so than MC-LR. For example, [33] found that MC-LA persisted throughout the recreational season (9.5 weeks) in a small temperate Canadian lake, long after the disappearance of surface cyanobacterial blooms (visible for five weeks), whereas MC-LR was found to decline much more rapidly (two to four weeks) in other lakes. MC-LA may also penetrate lipid membranes more readily (more hydrophobic) than MC-LR or -RR, increasing its likelihood of bioaccumulation, and it has been directly tied to wildlife mortalities [14]. MC-LA is also less readily removed by carbon filtration than MC-LR, -RR, and -YR [32], posing additional challenges for water treatment facilities. 

Compared to the large number of studies based on total microcystins, there are fewer studies on the dynamics of specific MC congeners, and these studies suggest there may be a wide range in congener composition among lakes [24,26]. Indeed, while MC-LA appears to be an important player in North American lakes, it seems less common in European lakes [7]. Though this difference may be due to differences in standards available for MC congener analysis, recent work suggests that regional differences in environmental conditions may explain some of the spatial variability in cyanotoxin composition. For instance, environmental factors such as water temperature, light regime, and thermal stratification were shown to be significant drivers of the cyanotoxin composition observed across a broad-scale synthesis of data from 137 European lakes [12]. These spatial patterns in cyanotoxin composition may admittedly reflect the dominance of certain cyanobacterial strains or species, which are themselves driven by environmental gradients (e.g., [28]). However, patterns in species composition may not necessarily determine patterns in MC congener composition. MCs are produced by a number of cyanobacteria (e.g., *Dolichospermum*, *Microcystis,* and *Planktothrix*), each of which can produce several MC congeners simultaneously [26,34,35,36]. In addition, there is a wide variation in congener composition among strains of the same species. Furthermore, given that guidelines associated with drinking water consumption are based on MC concentrations, and not species composition [12,37], and that MC congeners can vary in toxicity by one to two orders of magnitude [32,38], the questions of which MC congener dominates and why are important to resolve directly. Agencies would also benefit from robust predictions of impending toxic blooms and knowledge of which cyanotoxins are likely to occur under which routinely monitored environmental conditions [39]. Knowledge of the persistence of MC congeners in natural systems would also be important to reduce the discrepancy between lake closures and the period of potential toxin exposure [33,40]. Overall, a shift in focus from the analysis of one (MC-LR) to many (MC congener composition) will likely have important implications for the protection and management of freshwater resources.

In this study, we modelled changes in the relative concentration of different MC congeners (MC-LA, -LR, -RR, and -YR) in response to routinely monitored environmental factors using a raw water intake dataset collected by a single agency (the Ontario Ministry of the Environment, Conservation and Parks (OMECP)) using the same analytical methods. This entailed both spatial (19 intake sites situated across 12 main water bodies) and temporal (seven months sampled over 10 years) analyses to identify the importance of changes in meteorological conditions and lake nutrient concentrations on the variation of different MC congeners across the Great Lakes region of North America. We examined the potential role of weather-related variables including air temperature, precipitation, wind speed and wind direction as these have previously been shown to enhance cyanobacterial dominance and bloom formation [41,42,43]. We also examined the potential role of major nutrients (phosphorus (P) and nitrogen (N)) as they are strong predictors of cyanobacterial biomass, cyanobacterial dominance, and cyanotoxins in freshwaters [44,45]. Lastly, we considered characteristics of each intake site (depth and distance from shore) as well as dreissenid mussel control measures (raw water pre-chlorination) which may affect MCs [46,47]. To place this regional analysis within a global context, we conducted a synthesis of the peer-reviewed literature to test for systematic patterns of MC congener dominance across multiple regions. 

## 2. Results

### 2.1. Regional Analysis of the Great Lakes Intake Sites

The time period for which analytical data were available differed among MC congeners. Most were only quantified from 2013 onwards, MC-LR, -RR, and -YR were quantified from 2004 onwards, and MC-LA from 2006 onwards. We thus restricted our statistical analyses to years when the four most dominant congeners (MC-LA, -LR, -RR, and -YR) were measured (i.e., 2006–2016). We also restricted our analysis to ice-free months (April to November). This provided us with a regional raw water intake database spanning a 10-year period, collected during the open water season from 12 main water bodies in Ontario, Canada (Laurentian Great Lakes, Lake of the Woods, and other Ontario lakes and rivers, Figure 1). For the years and months examined, intake sites were sampled, on average (mean and median), four times per month (range: one to nine days per month), for a total of 2002 unique days-months-years-sites covering the years 2006–2016 and 19 intake sites (as some water bodies had multiple intake sites). 

Despite differences in the time of sampling across the main water bodies (e.g., complete sampling in Lake Ontario, intermittent in Detroit River and Lake Erie, and late-onset of sampling in Lake St. Clair (Figure 2)), we observed important spatial patterns in MC congener composition within the Great Lakes and the surrounding region. When averaging the concentrations of each congener across all sampling dates, we noted the variability in congener dominance among water sources (Appendix A). Some water bodies (e.g., Lake St. Clair) had greater MC-LA concentrations, while others (e.g., Lake Ontario) had greater MC-LR or MC-RR concentrations across time points. In general, MC-LA was higher at intake sites located along the Detroit River, Lake Erie, and Lake St. Clair, and was identified by the indicator value index (*indval*, Dufrêne and Legendre 1997) as having high fidelity and specificity to these sites (Figure 1), whereas MC-LR was an indicator of sites along Lake Ontario’s Bay of Quinte. These dominance patterns were also consistent from year to year within a given water body (Appendix A), notably so among the most routinely monitored sites with highest MC concentrations (i.e., intakes on Lake Ontario’s Bay of Quinte (Bayside, Belleville, Deseronto, and Picton), Lake Erie (Elgin, Essex, Pelee, and Union), Lake St. Clair (Lakeshore and Stoney Point) and the Detroit River). 

When examining all unique sampling dates (i.e., not averaged across years within a water body, nor across observations within a given year and water body) in the four most frequently monitored water bodies (Bay of Quinte, Lake Erie, Lake St. Clair and the Detroit River), we noted that the variance in MC-LA and -LR (the two dominant congeners), greatly increased after the year 2012 in the Bay of Quinte, with similar patterns in Lake Erie, Lake St. Clair and the Detroit River (Appendix A), though the sampling was more sporadic for the latter three prior to 2012. For the years monitored, the concentration of MC-LA likewise increased over time in Lakes Erie, St. Clair and the Detroit River (Appendix A). Consequently, the relative concentration of each MC congener changed over time (Figure 2), whereby MC-LA concentrations approached and even surpassed those of MC-LR and -RR after 2012. 

Environmental factors likewise varied over space and time in the four most frequently sampled water bodies of the Great Lakes basin. On the dates sampled, the Bay of Quinte region tended to experience warmer conditions than the other sites, whereas the Detroit River and Lake St. Clair region tended to experience higher wind and wetter conditions (Table 1). Meteorological conditions also varied over the years monitored and across stations (Appendix A). Lower wind speed, decreased precipitation, and higher maximum temperatures were recorded after 2012 at the meteorological station near the Detroit River and Lake St. Clair. Similar drops in wind speed and precipitation were recorded in 2015 and 2016 at the weather stations near Lake Erie and the Bay of Quinte (Appendix A). Across sites and time points, nutrient concentrations were typically within the oligo-mesotrophic range, with minimum, median, and maximum TP concentrations of 5, 21 and 675 µg/L, respectively. TP concentrations tended to be higher along the Bay of Quinte, whereas TN was highest at the Lake St. Clair intake sites (Table 2). There was a tendency for TP to decrease over time across the four most frequently monitored water bodies, whereas TN only decreased in the Bay of Quinte, Lake Erie and Lake St. Clair sites (Appendix A). No significant trend in TN was detected in the Detroit River. The depth of the intake sites, their distance from shore, and the use of pre-chlorination to control for dreissenid mussels (pre-chlorination in 59% of sites) also differed among intake sites and main water bodies (Appendix A, Table 3). 

As mentioned previously (and shown in Appendix A), most water bodies were only intermittently monitored. To thus provide a more robust investigation of the distribution pattern of MC congeners in time and space, we restricted further statistical analyses to the four water bodies most frequently sampled by the OMECP (Bay of Quinte, Lakes Erie and St. Clair, and the Detroit River). In terms of relationships among MC congener composition and environmental change, a multivariate canonical ordination (redundancy analysis, RDA) showed that the relative abundance of each congener varied with meteorological conditions, nutrient concentrations, chlorine treatment (Yes/No), and the distance of the intake sites from shore (Figure 3). The relative abundance of MC-LR and -RR increased as mean monthly maximum temperatures and TP increased, and as the direction of the wind with the highest speed changed from southwesterly winds (200°) to northern winds (360°). In contrast, the relative abundance of MC-LA increased as maximum wind speeds decreased. Furthermore, intakes closer to shore tended to have higher MC-LR and -RR, and chlorinated sites tended to have higher MC-LA (Figure 3). The multivariate linear model examined with this RDA accounted for a small portion of the total variance in MC congener composition (R^2^-adj = 0.09). We also identified a clustering among observations from the same water body (Figure 3) and including water body as a co-variate in the RDA accounted for an additional 7% of the variance in MC composition (figure not shown). A limitation of the RDA, however, is that observations with missing environmental data were omitted prior to analysis and that the relationships were based on linear regressions.

The use of a multivariate regression tree (MRT) analysis helped identify significant non-linear responses to environmental factors, as well as their interactions, which together accounted for an additional 25% of the variance in MC congener composition (R^2^ = 0.34, Figure 4). Among the variables tested, wind speed was the most important explanatory variable in predicting congener composition (Figure 4, tree node 1), but temperature, precipitation, and nutrients had significant secondary effects. When average monthly winds were very stable (maximum wind speed <36 km/hr on average), MC-LR dominated, followed by MC-RR (Figure 4, node 5). When the wind was relatively stable (maximum wind speed between 36 and 51 km/hr) coupled with low average monthly precipitation (<1.6 mm), MC-LR still dominated, followed by MC-RR (Figure 4, node 5). In contrast, when intermediate winds (36 and 51 km/hr) were coupled with precipitation above 1.6 mm, MC-LA was the main MC encountered, with more MC-LA than other congeners when temperatures were cooler (<13 °C). Under wet but warmer conditions (≥13 °C), MC-LA, -LR, and -RR were generally more abundant when TP was at least in the mesotrophic range (TP > 8.5 µg/L) (Figure 4, node 8), but MC-LA was greatest when TN was below 169 µg/L (Appendix A). Under high wind speeds (maximum wind speed > 51 km/hr), combined with warm air temperatures (MAT > 14 °C on average) and as the direction of wind with the highest speed changed from southwest to more northern winds, MC-LR once again dominated, especially under nutrient-rich conditions (TP > 26 µg/L), followed by MC-RR (Figure 4, node 4). 

To tease apart potential effects of geographical location from those of the landscape-scale environmental gradients, we used univariate mixed-effect regression trees with either the concentration of MC-LA or -LR, the two most common congeners detected in the region, as the response variables (the analysis provided similar results when using the relative abundance of MC congeners instead of absolute concentrations, and although not shown MC-RR behaved as MC-LR). The univariate trees showed that MC-LR concentrations were highest under warm, high nutrient conditions (Figure 5a) and that MC-LR concentrations were higher on average in the Bay of Quinte (Figure 5b). In contrast, MC-LA concentrations were higher on average in Lake St. Clair, and primarily related to wind speed (Figure 5c,d). The concentrations of both MC congeners, where greatest from 2013 onwards (Figure 5). 

### 2.2. Global Analysis of the Peer-Reviewed Literature

At the spatial scale of the Great Lakes basin and the surrounding region, we detected significant heterogeneity in MC congener dominance (notably between MC-LA and MC-LR, -RR) due in part to environmental variability among water bodies and over time. Within a global context, the synthesis of studies reporting MC congener data likewise showed a pattern in MC congener occurrence. In particular, MC-LA was most common in North and South American lakes (Figure 6a, Kruskal–Wallis test: χ^2^ = 44.76, *p* < 0.0001). The lakes sampled in this continent (predominantly the US and Canada) tended to have a smaller surface area, be shallower in depth, and have lower TP concentrations than the lakes sampled in the other five continents (Table 3, Appendix A). MC-LR was more evenly distributed (Figure 6b, Kruskal–Wallis test: χ^2^ = 1.73 *p* = 0.943), though most common in eutrophic lakes (i.e., intermediate TP range for these sites (Appendix A)), which corresponded to lakes sampled in Africa, Asia, Europe, and North America (Appendix A, Table 3). Lastly, MC-RR and -YR tended to divide the depth niche space (the former being relatively more abundant in lakes of intermediate depth, the latter being relatively more abundant in deeper lakes (Appendix A)). 

## 3. Discussion

### 3.1. Regional Relationship between Congener Occurrence and Environmental Conditions 

Our regional analysis of the Laurentian Great Lakes showed that the variability in MC congener composition across raw water intake sites of Southern Ontario, Canada could be due to differences in meteorological conditions and lake nutrient status. MC-LA and MC-LR were the most commonly observed MC congeners in the region, though their dominance was distinct across the landscape (Figure 1). As environmental conditions changed (lower wind speed, decreased precipitation and higher temperatures in most recent years (Appendix A)), the relative abundance of MC-LA and MC-LR also changed (increase proportion of -LA in western locations (Figure 2)), and the concentration of both MC congeners became increasingly variable among sampling dates from 2013 onwards (Appendix A). This observation led us to further assess whether environmental factors influenced MC congener prevalence. Our findings showed that when intermediate winds (average monthly wind speeds ranging from 36 to 51 km/hr) were coupled with wetter conditions, MC-LA tended to dominate. These conditions were typical of the raw water intake sites in Lake St. Clair (Figure 5c,d), as well as along the Detroit River and Lake Erie (Figure 1). In contrast, either weak wind (<36 km/hr), or stronger winds (>51 km/hr) coupled with warm conditions (>14 °C) and nutrient-rich waters (TP > 26 µg/L) were related to the dominance of MC-LR (and co-dominance of MC-RR). These conditions were typical of the Bay of Quinte (Figure 1 and Figure 5a,b). 

The overriding effect of monthly-averaged wind speed on MC composition in this regional dataset is noteworthy (Figure 4). Wind speed, through changes in turbulent mixing and water temperature, is known to affect phytoplankton and cyanobacteria species composition due to differences in buoyancy regulation and temperature optima [48]. That is, gas-vacuolated species can regulate their buoyancy and overcome settling to optimize resource acquisition during low wind speeds, strong stratification and high irradiance [41,49]. Michalak et al. [50] found that the combined effect of calm wind conditions, reduced lake mixing, increased nutrient loading and increased precipitation may have facilitated a record-breaking *Microcystis* bloom in Lake Erie in 2011, and furthermore, predicted increases in the frequency of such events in the future (50% increase in large storms with precipitation >30 mm under future climate models). Recently, Kelly et al. [51], found that lower wind speed (≤37 km/hr), combined with an increase in eutrophication indicators (Chl *a*, nutrients) and temperature, were associated with the increased probability of total MC concentrations exceeding drinking water standards (1.5 µg/L) in the Bay of Quinte. Given the decline in wind speed observed in recent years in the Laurentian Great Lakes region (Appendix A) and predictions of higher total annual precipitation [52], the increase in MC-LA dominating blooms may very well continue. However, MC-LA dominance was also observed in a small Ontario lake with no flushing (i.e., a closed-basin lake with no inflow or outflow [33]). Thus, the cumulative effect of environmental stressors (wind, precipitation, and nutrients) may vary across the landscape and interact with other factors such as lake morphometry. Furthermore, threshold wind speeds for complete mixing of the water column are likely to vary among water bodies and regions. In some lakes, winds greater than 29 km/hr were required to favour non-buoyant species [53,54], whereas other systems required stronger winds (>72 km/hr, [41]).

The likely reason for the observed relationship between environmental factors and MC congener dominance is that different congeners are produced by different cyanobacterial species or strains, which themselves vary along their ecological niche [55]. MC congener composition has been shown to vary between and within species (i.e., among strains), and to a lesser extent within strains. For example, within-strain variation in congener composition has been observed in response to changes in temperature [56], light availability [57,58] and nitrogen concentrations [59,60]. Different species may also produce many MC congeners concurrently. For instance, *Planktothrix* blooms have been dominated by both MC-LA (North American lakes [61]) and the -RR variant (European lakes [62,63,64,65]). Similarly, *Microcystis aeruginosa* has been associated with MC-LA dominance in some lakes (North America [33]) but -LR dominance in others (European lakes [66]). 

A caveat of this regional study is the limitation of raw water intake data in terms of extrapolating to the surface water conditions. There may be times of the year (e.g., during storm events) when toxins measured at intake sites are partially or wholly derived from resuspended sediments, and may be less representative of surface water conditions. This is especially an issue given that MC-LA is more resilient to degradation and has lower sediment adsorption than MC-LR [32,67]. The differential resuspension and removal efficiency of MC congeners may thus vary with the depth of intake (greater exchange between surface and bottom waters in shallow sites). For the sites examined here however, we failed to detect any significant effect of depth of intake on MC congener composition, and all intake sites were relatively shallow (<12 m). Instead, we found that the distance from shore was more important (Figure 2), which may be related to the difference in water currents. 

Chlorine treatment and dreissenid mussel occurrence could further decouple MC congener composition between surface and intake waters. Indeed, many water treatment plants chlorinate the raw water to serve as a chemical barrier to prevent dreissenid mussel veligers from clogging up intakes pipes as well as to prevent them from moving into the water treatment plant infrastructure. In this regional analysis, we detected a weak effect of chlorination on MC congener composition, which may be driven by the secondary effects of zebra mussel presence. There are reports, for instance in the US Midwest, of higher MC levels in dreissenid-infested water bodies compared to those without mussel [47]. This, in turn, may be linked to the changes in nutrient ratios and concentrations created by the mussels towards N:P ratios more suitable for toxic cyanobacteria growth [47,68,69,70,71]. Interestingly, distance from shore and chlorination were not selected by the MRT analysis. Wind speed and air temperature (>12 °C) were selected instead, though many water treatment plants only chlorinate when water temperatures exceeded 12 °C (i.e., during increased zebra mussel reproduction). Furthermore, although the effect of distance from shore and chlorine treatment on MC congener composition was less clear (Appendix A), the concentration of total MC within the four most frequently monitored water bodies tended to decrease with depth and distance from shoreline (notably so among the intake sites along Lakes Ontario and Erie (Appendix A)) and increase with chlorination (or presence of zebra mussels Appendix A). From a water treatment perspective, we suggest that more work is needed to determine the effect of chlorination and intake location on total MC concentration and on the relative composition of MC congeners, which may guide treatment optimization (such as using additional treatment methods at MC-LA dominated sites). 

### 3.2. Global Relationships between Congener Occurrence and Environmental Conditions

Our meta-analysis of the peer-reviewed literature on MC congener dynamics identified significant differences in dominance patterns among continents. In general, the relative proportion of MC-LA was low for the lakes sampled in Africa, Europe, and Asia, but notably greater in lakes sampled in North and South America (Figure 6a). In contrast, MC-LR was more uniformly observed across sample sites (Figure 6b). In light of the relationships identified at the regional scale (Great Lakes and the surrounding region), we explored whether the difference in MC-LA occurrence among continents and countries was due in part to differences in environmental conditions among locations. We noted that climate (wind speed) was the most important driver of MC congener composition in the Great Lakes regions, and concordantly, MC-LA was more common in lakes with smaller surface area (i.e., North and South American lakes (Figure 6a, Appendix A)), which may modulate climate signals such as wind exposure [72,73]. We also noted that the effect of nutrients on regional congener composition was weaker, but still an important driver of MC-LR, with the latter more common at higher nutrient concentrations (Figure 5a). Similarly, on the global scale, we found that MC-LR was more common in eutrophic lakes (Appendix A). In general, despite the lack of a standardised protocol among the studies synthesizes in our meta-analysis, the global-scale relationships between lake morphometry, nutrient concentrations, and MC congener composition in the water column echoed the regional patterns observed in the Great Lakes Basin raw water intake data.

## 4. Conclusions

At the two geographical scales (regional and global) examined here, we identified strong spatial heterogeneity in congener composition, explained in part by routinely monitored environmental conditions. In particular, we found that:In the Laurentian Great Lakes, MC-LA was often more prevalent than MC-LR, and its concentration has increased over the last decade at several sites.The more toxic congeners (MC-LA and -LR) occurred under both low nutrient (MC-LA) and high nutrient (MC-LR) concentrations, while meteorological conditions (wind speed and precipitation) determined the relative concentration of each.Meso-oligotrophic waters with intermediate winds and frequent rain events showed greater percentage of MC-LA, while low winds or high winds combined with warm, nutrient-rich conditions showed greater percentage of MC-LR and -RR.Environmental conditions and related MC congener dominance were geographically distinct, with conditions that favoured MC-LA in the western part of our regional study (Lake St. Clair, Detroit River, and parts of Lake Erie), but windy, warm, eutrophic conditions that favoured MC-LR observed to the east in Lake Ontario’s Bay of Quinte.Globally, MC-LA and MC-LR also showed geographically distinct patterns, with a smaller percentage of MC-LA in Africa, Europe and Asia, compared to North and South America. These patterns of MC congener dominance were associated with differences in lake morphometry and nutrient concentrations: MC-LA tended to be more prevalent in smaller lakes while MC-LR peaked in eutrophic lakes.

## 5. Materials and Methods 

### 5.1. Great Lakes Regional Analysis 

#### 5.1.1. Analytical Data

Raw water samples were collected by water treatment plant personnel and sent to the Ontario Ministry of the Environment, Conservation and Parks (OMECP) for chemical analysis. For each raw water sample, the concentration of total MC (free and intracellular) was quantified by the OMECP laboratory technicians by isolation on silica gel and analysis by liquid chromatography-electrospray ionisation tandem mass spectrometry [74]. The concentrations of MC congeners (MC-LR, -RR, -LA, -YR, -LY, -WR, -HtyR, -HilR, -LW, -LF, and desmethyl-MC-LR, desmethyl-MC-RR) were measured using an internal standard quantification with multi-point calibration approach [74]. 

#### 5.1.2. Statistical Analysis

The environmental factors considered as potential drivers of congener composition were meteorological condition (monthly averages of total precipitation (mm), wind direction (tens of degrees), wind speed (km/hr), and minimum, maximum and mean air temperature (°C) from nearby weather stations, Canadian Daily Climate Database) and nutrient concentrations (total phosphorus (TP) and total nitrogen (TN) from the raw water samples collected for MC analysis). Due to the proximity of Lake St. Clair and the Detroit River raw water intake sites (Figure 1), we used the same meteorological station for both water sources. We also considered the depth and distance from shore of the intake sites as these variables may track differences in water currents, light, and nutrient availability. Finally, we tested for any detectable effect of raw water pre-chlorination (used for dreissenid mussel control) as this may affect the concentration of total MC and MC congener composition (both via the direct effect of chlorine on the cyclic peptide structure and indirectly via secondary effects on dreissenid mussels [46,47]). Although the total chlorine concentration in raw water samples was measured by the OMECP, concentrations were often below detection (only detected in sites along Lake Ontario, Lake Erie, and the Detroit River (Appendix A)). To provide chlorine treatment data for all sample dates and sites, we thus used a categorical classification to indicate whether chlorination treatment was confirmed by the water treatment plants (Yes, No or turned Off prior to sample collection (Appendix A)). This classification was an accurate indication of chlorine concentrations when detected (i.e., in Lake Ontario, Lake Erie, and the Detroit River (Appendix A)). To create a variable that more broadly indicated chlorination treatment as well as the presence of dreissenid mussels, we grouped the one site where chlorination was turned off prior to sample collection (Union Water Treatment Plant along Lake Erie) with sites having continued chlorination.

To examine how the concentrations of all four congeners (MC-LR, -RR, -LA, and -YR) varied with changes in the environmental factors over time and space, we conducted a multivariate canonical ordination (redundancy analysis (RDA)) using the *rda* function from the {vegan} package in R [75]. To test for changes in the relative concentrations of MC congeners, we transformed the response matrix into relative abundances using the argument “total” of the *decostand* function. Although RDA examines the relationship between the multivariate response matrix and the suite of environmental factors, relationships are restricted to linear regressions. In addition, the method does not allow for missing data. We thus removed observations with missing environmental data prior to running the RDA. 

Given that the RDA only tests for linear relationships between the multivariate response matrix and environmental drivers, we coupled this analysis with a multivariate regression tree (MRT, function *mvpart* in R [76,77]), which allows for non-linear and threshold responses [78]. In addition, MRTs allow for missing data. We used the cross-validation relative error (CVRE), which is the ratio of the variation unexplained by the tree to the total variation in the response, as the criterion for selecting the most parsimonious tree (i.e., the tree with the least splits whose CVRE value is within one standard error of the tree with the lowest CVRE [79]). 

To further test whether similarity in meteorological and nutrient parameters among sites of closer proximity (Figure 1) could lead to the dominance of any particular MC congener, we ran a linear mixed effect regression tree (using the {glmertree} package in R [80]) with a random effect for year and site (main water body). By teasing apart the effect of co-location, this additional analysis evaluated whether the relationships observed with the multivariate regression tree on all sites and years were biased by the sampling design.

Lastly, to identify which congener was most often associated with which water body, we conducted an indicator species analysis using the indicator value index (function *indval* of the {labdsv} package [81]). Briefly, *indval* measures the fidelity and specificity of a “species” (congener) to a group (water body). Where specificity is defined as the mean abundance of the species within the targeted group compared to its mean abundance across all groups, and fidelity is the proportion of sites of the targeted group where the species is present [82]. The index is thus maximized when a species (congener) is observed at all sites belonging to a single group (water body), and not elsewhere [76].

### 5.2. Global Analysis

To assess the patterns in MC congener composition across many regions, we conducted a synthesis of published literature that provided MC congeners concentration data from freshwater lakes and reservoirs. Specifically, we conducted an ISI Web of Science and Google Scholar search using the keywords “microcystin”, “congener”, “cyanotoxin”, “cyanobacteria”, “lake”, “eutrophication”, “microcystin-LA”, “MC-LA”, “microcystin-LR” and/or “MC-LR” published between 2000 and 2017. We screened the studies using the criteria that: 1) Concentrations of MC congeners were measured (i.e., not just total MCs), 2) MC-LR, -LA, -RR and -YR were quantified (i.e., the use of the relevant standards was mentioned), and 3) data tables or graphs from which the raw data could be digitized were provided in each publication. A total of 146 sites which covered all continents (to the exception of Antarctica) met these search criteria (Appendix A). Most studies presented their data as concentrations in the water column (µg/L), while some presented their data as seston content (g/dry weight). To analyze both types of concentration data, we calculated the composition as percentages of total MCs reported.

## Figures and Tables

**Figure 1 toxins-11-00620-f001:**
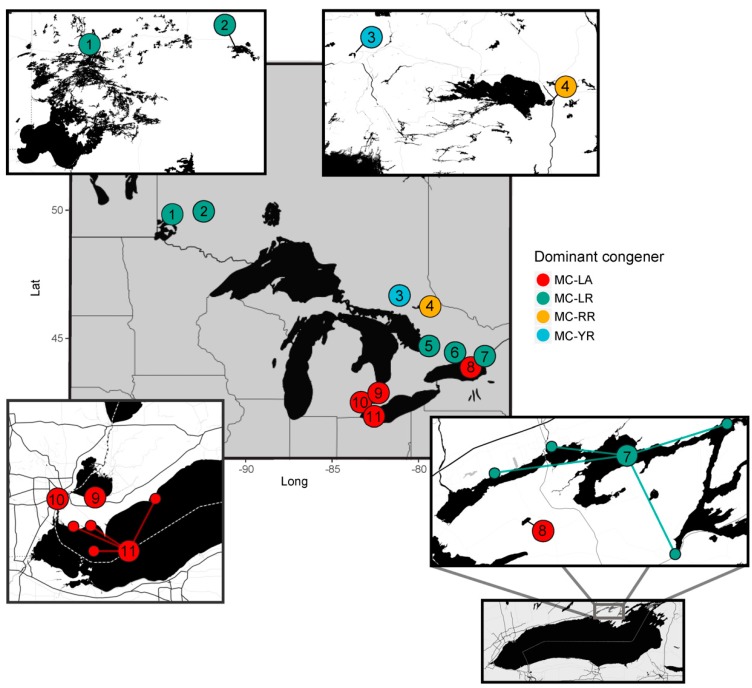
Map of main water bodies with raw water intake samples analyzed by the OMECP. Points are colour coded by dominant congener as determined by an indicator analysis (*indval* function from the {labdsv} package in R), which showed that Robin Lake (**8**), Lake St. Clair (**9**), Detroit River (**10**) and Lake Erie (**11**) were dominated by MC-LA (red), Lake of the Woods (**1**), Lake Wabigoon (**2**), Lake Couchiching (**5**), the Otonabee River (**6**), and Lake Ontario’s Bay of Quinte (**7**) were dominated by MC-LR (green), Ramsey Lake (**3**) was dominated by MC-YR (blue), and Lake Nipissing (**4**) by MC-RR (orange). Note: There are four intake sites located along Lake Erie (Elgin, Essex, Union, and Pelee island), four along the Bay of Quinte (Bayside, Belleville, Deseronto, and Picton) and two along Lake St. Clair (Lakeshore and Stoney Point). All other main water bodies only have one intake site.

**Figure 2 toxins-11-00620-f002:**
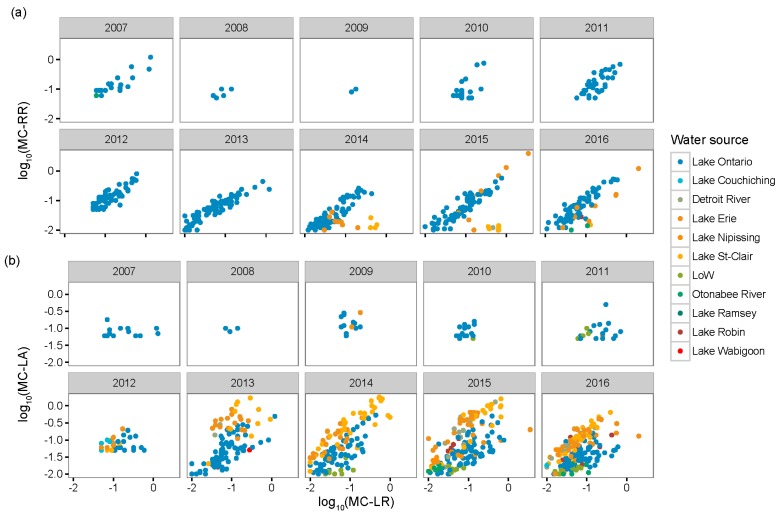
Relationships between dominant congeners from the Laurentian Great Lakes basin raw water intake data. Log-transformed concentrations of (**a**) MC-RR and (**b**) MC-LA vs. MC-LR.

**Figure 3 toxins-11-00620-f003:**
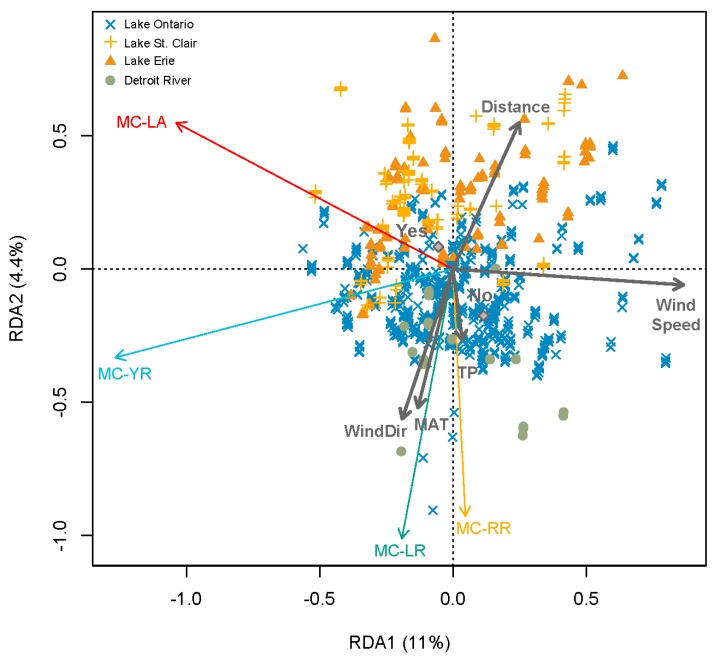
Redundancy analysis of the relationships between the relative abundance of microcystin congeners (i.e., MCL-A, -LR, -RR, and -YR) and environmental factors (i.e., meteorological conditions, nutrient concentrations, intake location, and chlorine treatment) from the most frequently sampled water bodies of the Laurentian Great Lakes basin raw water intake data. Quantitative environmental factors were scaled and centered to reduce variance prior to analysis. The centroids of the qualitative variable (categorical variable for chlorine treatment, Yes/No) are shown by the grey diamonds.

**Figure 4 toxins-11-00620-f004:**
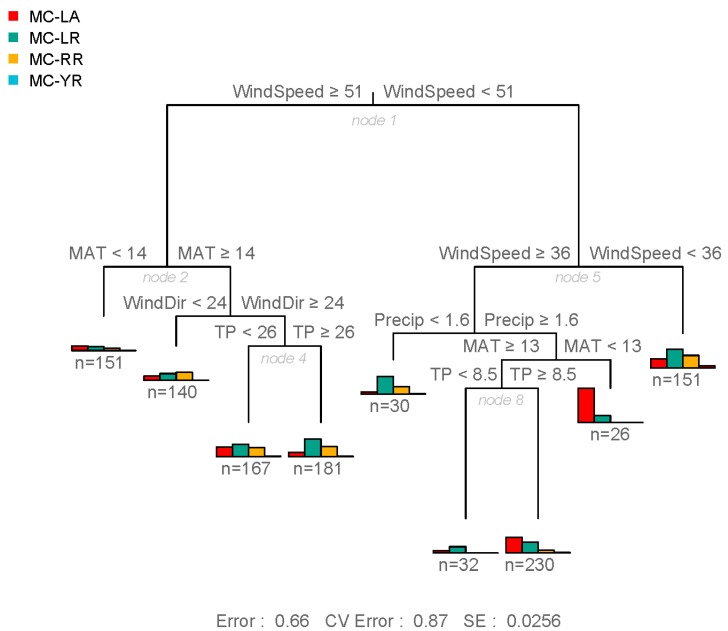
Multivariate regression tree (MRT) of congener relative abundance from the most frequently sampled water bodies of the Laurentian Great Lakes basin, constrained by meteorological conditions (MAT = maximum temperature, WindSpeed = maximum wind speed, WindDir = direction of wind with the highest speed, Precip = precipitation) and nutrient concentrations (TP = total phosphorus).

**Figure 5 toxins-11-00620-f005:**
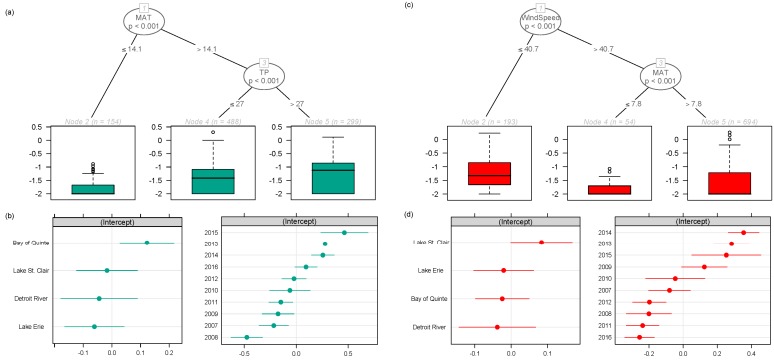
Mixed effect regression tree analysis for MC-LR and MC-LA concentrations from the most frequently sampled water bodies of the Laurentian Great Lakes basin, where fixed effects relationships are shown in (**a**,**c**) and random intercept coefficients are shown in (**b**,**d**) for MC-LR and -LA, respectively. MAT = monthly averaged maximum air temperature (°C), TP = Total Phosphorus (µg/L), and WindSpeed = monthly averaged maximum wind speed (km/hr).

**Figure 6 toxins-11-00620-f006:**
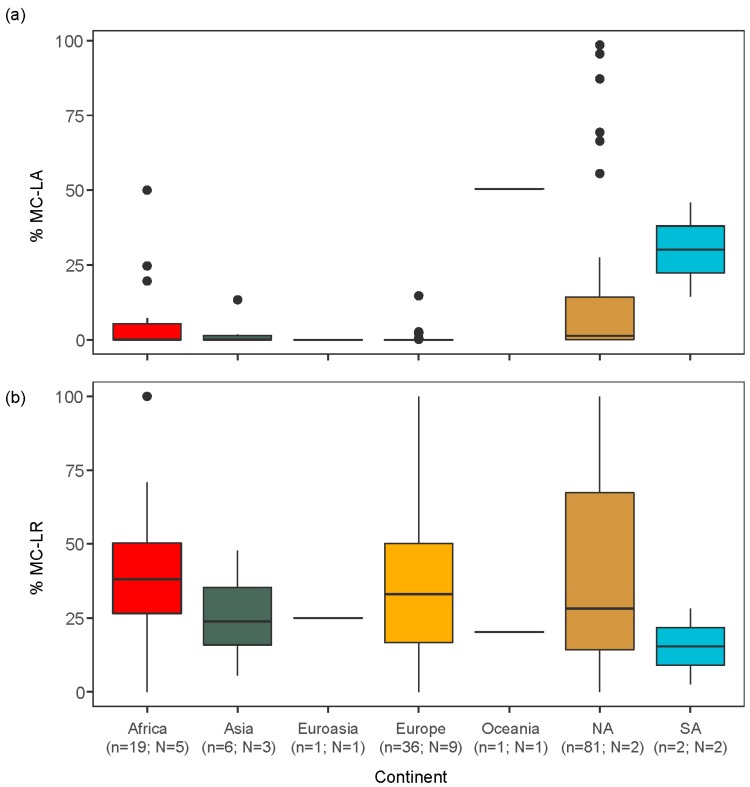
Summary of global meta-analysis of the peer-reviewed literature (OMECP raw intake sites were omitted). Boxplot of the percentage of (**a**) MC-LA and (**b**) MC-LR in different continents, where n = number of lakes and N = number of countries in each continent.

**Table 1 toxins-11-00620-t001:** Summary statistics of the climate data (monthly averages from April to November, for the 2006–2016) from weather stations near the four main water bodies most frequently monitored by the Ontario Ministry of the Environment, Conservation and Parks (OMECP).

Environmental Factor	Percentile	Stn near L Ontario (*N* = 456)	Stn near L St. Clair and Detroit R (*N* = 236 & 118)	Stn near L Erie (*N* = 476)
MAT	min	−7.5	−9.3	−13.7
	5th	−2.6	−4.4	−9.2
	35th	8.8	6.7	2.3
	50th	14.3	13.0	10.3
	75th	23.5	20.9	21.4
	95th	27.4	24.8	25.5
	max	28.8	26.5	27.5
Precipitation	min	0.8	0.8	0.6
	5th	1.3	1.9	1.4
	35th	48.1	54.1	5.5
	50th	70.5	73.8	42.6
	75th	98.0	110.2	73.2
	95th	125.3	166.3	122.7
	max	195.6	207.0	150.0
Wind direction	min	2.0	1.0	1.0
	5th	4.0	7.2	2.8
	35th	24.0	21.0	19.0
	50th	25.0	23.0	20.5
	75th	28.0	26.0	22.8
	95th	31.0	29.0	34.3
	max	35.0	34.0	36.0
Wind speed	min	32.3	36.1	36.2
	5th	33.7	37.3	39.3
	35th	52.5	52.0	45.1
	50th	57.0	60.0	51.0
	75th	69.0	74.0	65.0
	95th	91.0	95.3	78.5
	max	100.0	111.0	100.0

MAT = maximum air temperature, and *N* = total sample size (only one station used for each water body, to the exception of Lake St. Clair and Detroit River where the same station was used due to proximity of both water bodies). Due to some NAs, *N* = 384 observations for wind direction in Lake Erie, whereas all other climate variables have *N* = 476 observations in Lake Erie.

**Table 2 toxins-11-00620-t002:** Summary statistics of nutrient data (months: April–November, years: 2006–2016) from the four main water bodies most frequently monitored by OMECP.

Water Source	Percentile	L Ontario (*N* = 771 *n sites* = 4)	L St. Clair (*N* = 128 *n sites* = 2)	Detroit R (*N* = 30 *n site* = 1)	L Erie (*N* = 195 *n sites* = 4)
TN	min	25.0	79.0	112.0	104.0
	5th	180.2	209.0	139.6	170.6
	35th	460.0	390.0	320.2	330.0
	50th	510.0	490.0	345.0	416.5
	75th	610.0	888.0	413.0	576.0
	95th	869.8	1982.0	558.8	730.9
	max	2896.0	4020.0	1340.0	1244.0
TP	min	6.0	6.0	6.0	5.0
	5th	12.0	10.0	7.0	7.0
	35th	22.0	15.0	11.0	12.0
	50th	27.0	18.0	12.0	14.0
	75th	39.0	25.0	15.5	19.0
	95th	63.0	36.0	28.5	42.2
	max	675.0	66.0	179.0	139.0

TN = total nitrogen (µg/L), TP = total phosphorus (µg/L), *N* = total sample size, and *n sites* = number of raw water intake sites within each main water body.

**Table 3 toxins-11-00620-t003:** Summary of morphometric and nutrient (total phosphorus (TP)) variables for the Laurentian Great Lakes basin (raw water intake sites analyzed by OMECP) and global synthesis of the literature. *N* represents the number of main water bodies.

	Great Lakes Basin Dataset			Global Dataset					
	All Intake Sites			N & S America			Africa, Asia, Europe & Oceana		
	Depth at Intake (m)	Distance from Shore (m)	TP (µg/L)	Surface Area (ha)	Maximum Depth (m)	TP (µg/L)	Surface Area (ha)	Maximum Depth (m)	TP (µg/L)
Range	0–12	0–1300	5–675	4.38–72,500	1–44	2–2047	0.7–305,000	1–175	2–14,420
Median	5	340	20	127	5.4	41.5	521	7.4	69
Mean	5.1	320	27	1633	7.7	150	11065	17	700.7
*N*	17	16	17	65	69	60	108	110	95

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
