# Peer review of "Meteorological and Nutrient Conditions Influence Microcystin Congeners in Freshwaters"

_toxins, 2019, doi:10.3390/toxins11110620_

Round 1
Reviewer 1 Report
This study aims to address the important subject of the heterogeneity in the abundance of microcystin congeners in bodies of water from different regions of the Great Lakes basin and as well as a global review of lake data from the literature. To approach this topic the authors impressively utilize a data set that spans a 10-year period gathered by the same agency and using the same protocols. The authors clearly describe the limitations to their study and the conclusions are strongly supported by the data presented. There are only a few minor revisions that I suggest.
Minor revisions
There is a typo in the beginning of the fourth paragraph of the introduction (beginning on line 90). “In this this study,”. Table 2 was mentioned before Table 1. Please present tables in order as they appear in the manuscript. Is there a reason that Lake Wabigoon is missing from Figure S2? Although depth is discussed and the study is constrained to ice-free months, I am curious about the impact that the colder months may have (variations in time spent iced-over, minimum temperature reached, etc.).Author Response
We thank the reviewer for these comments and corrections. Please see our responses to the concerns raised below:
1) We have corrected the typo in the beginning of the fourth paragraph of the introduction. The start of line 90 now reads “In this study,” as opposed to "In this this study,".
2) We have switched the orders of Tables 1 and 2 (moving what was Table 2 up, and what was Table 1 down) so that they are presented in order of their first appearance in the manuscript.
3) Our apologies for this oversight. Lake Wabigoon was omitted from Figure S2 as it only had data for one year (thus the barplot for this lake was the same as that presented in Figure S1). To avoid redundancy and save on space, we had thus opted to omit this lake from Figure S2. We should have specified this important point in the figure caption. The new version of the Figure S2 caption explains why Lake Wabigoon was omitted from the figure.
4) Regarding examining the impact that the colder months may had on MC composition, we agree that this would have made for an interesting study. Unfortunately, the Ministry which provided this data only sampled for MC during the open water season.
Reviewer 2 Report
The manuscripts presents the results of an interesting research study investigating the distribution pattern of microcystin (MC) congeners in time and space, and the relationships between their occurrence vs. meteorological and nutrient conditions (total phosphorus and nitrogen concentrations determined in raw water samples collected for MC analysis). A regional raw water intake database spanning a ten-year period (2006-2016) was created, which contained samples collected from twelve main water bodies in Ontario, Canada. The analyses were performed based on the data provided by the Ontario Ministry of the Environment, Conservation and Parks (raw water samples were collected by water treatment plant personnel, and the concentrations of total MC and anatoxin-A in water were determined by liquid chromatography-mass spectrometry). Statistical analyses were performed using procedures from the package in R, and dedicated ordination and classification tests; in my opinion, appropriate methods were selected for statistical data processing. Global data were analyzed based on a synthesis of peer-reviewed literature.
Comments:
According to the Results section, a total of 2002 days-months-years-sites covering the years 2006-2016 were analyzed. However, the Authors also stated that only a portion of the data contained in the database could be used in the analyses - according to Figure 4, the multivariate regression tree was constructed using approximately half of the dataset (N=1108), but the relevant information is not provided in the main text. Another important consideration is the representativeness of data in spatio-temporal gradients (spatial and temporal uniformity), which is difficult to evaluate since an unknown amount of missing environmental data were omitted in analyses (cf. e.g. line 192).
Total sample size (N) and number of sites (n) should be provided in Table 2.
Despite the above concern regarding data representativeness, in my opinion the classification and ordinance analyses were based on sufficient datasets, and an analysis of changes in the studied parameters (gradients) delivered interesting results, although (as stated in the Discussion section) the values of these parameters were determined for selected water bodies in Canada and should not be directly extrapolated to other areas. Nevertheless, the relationships between meteorological conditions, Tot-P and Tot-N concentrations in water and MC structure in lakes provide novel and valuable insights into the communities of toxin-producing cyanobacteria in freshwaters. Another important research finding is that there exist two distinct patterns of MC dominance, which differ between America and Europe. The results are interestingly interpreted in the Discussion section, indicating the need for further research.
Author Response
We apologize for the lack of clarity on which data were summarized in each figure presented in the manuscript. Our goal was to first provide a global overview of all waterbodies (using maps, scatterplots and bar plots; Figures 1, 2, S1 and S2) to highlight, in part, that four waterbodies were more frequently sampled that others, and thus better suited for the spatio-temporal statistical analyses that followed.
We now more clearly indicate the lakes selected on
lines 144-145:
"When examining all unique sampling dates (i.e., not averaged across years within a waterbody, nor across observations within a given year and waterbody) in the four most frequently monitored waterbodies (Bay of Quinte, Lake Erie, Lake St. Clair and the Detroit River), ..."
lines 156-157:
"Environmental factors likewise varied over space and time in the four most frequently sampled waterbodies of the Great Lakes basin."
and lines 187-190:
"As highlighted above (e.g. Fig. S2), most waterbodies were only intermittently monitored. To thus provide a more robust investigation of the distribution pattern of MC congeners in time and space, we restricted further statistical analyses to the to four waterbodies most frequently sampled by the OMECP (Bay of Quinte, Lakes Erie and St. Clair and the Detroit River)."
We also provide the N (number of observations) for the table summarizing the meteorological data for the four main waterbodies. Please note that this is now Table 1. We do not provide n (number of sites) as only one weather station was used for each waterbody (to the exception of St. Clair and Detroit R. where the same weather station was used due to the proximity of both waterbodies).
Reviewer 3 Report
The manuscript with the title “Meteorological and nutrient conditions influence 2 microcystin congeners in freshwaters” present a detailed research on occurrence of different microcystins from sites around the Great Lakes. This time and space resolved samples combined with meteorological and data were properly analyzed and presented. Some interesting conclusions were made by the authors with are supported by the data. Thus, I recommend this manuscript for publication.
Author Response
We thank the reviewer greatly for their support and recommendation to publish our manuscript.